# Energy landscape underlying spontaneous insertion and folding of an alpha-helical transmembrane protein into a bilayer

Wei Lu[1,2], Nicholas P. Schafer[1,3] & Peter G. Wolynes[1,2,3,4]

Membrane protein folding mechanisms and rates are notoriously hard to determine. A recent force spectroscopy study of the folding of an $\alpha$-helical membrane protein, GlpG, showed that the folded state has a very high kinetic stability and a relatively low thermodynamic stability. Here, we simulate the spontaneous insertion and folding of GlpG into a bilayer. An energy landscape analysis of the simulations suggests that GlpG folds via sequential insertion of helical hairpins. The rate-limiting step involves simultaneous insertion and folding of the final helical hairpin. The striking features of GlpG's experimentally measured landscape can therefore be explained by a partially inserted metastable state, which leads us to a reinterpretation of the rates measured by force spectroscopy. Our results are consistent with the helical hairpin hypothesis but call into question the two-stage model of membrane protein folding as a general description of folding mechanisms in the presence of bilayers.

[1] Center for Theoretical Biological Physics, Rice University, Houston 77005 TX, USA. [2] Department of Physics, Rice University, Houston 77005 TX, USA. [3] Department of Chemistry, Rice University, Houston 77005 TX, USA. [4] Department of Biosciences, Rice University, Houston 77005 TX, USA. These authors contributed equally: Wei Lu, Nicholas P. Schafer. Correspondence and requests for materials should be addressed to P.G.W. (email: pwolynes@rice.edu)

Transmembrane proteins mediate crucial biological processes, including signaling across membranes, selective transmission of molecules through membranes, and proteolysis of proteins embedded in membranes. The sequences and structures of many transmembrane proteins are now known. The biophysical tools for characterizing the folding and stability of transmembrane proteins, however, are limited in comparison to those available for studying soluble proteins. In the case of soluble proteins, these tools have been crucial in illuminating folding mechanisms[1,2]. Experimental assays developed for use on soluble proteins are often not straightforwardly applicable to membrane proteins because bulk experiments on membrane proteins require the presence of a bilayer or bilayer-mimicking environment to keep the highly hydrophobic transmembrane proteins soluble. The presence of the bilayer complicates both the application of optical spectroscopic techniques and the modulation of the equilibrium between the folded and unfolded states. Ultimately, the lack of tools for measuring stabilities and kinetics of membrane proteins has shrouded in mystery the detailed folding mechanisms of membrane proteins in their natural bilayer-like environments. In this work, we compute the energy landscape underlying spontaneous insertion and folding of a multipass $\alpha$-helical transmembrane protein, GlpG, and thereby gain detailed structural insight into its folding mechanism in the presence of a bilayer. We validate our results by carrying out a critical comparison of our computed landscape with the results of single-molecule experiments of Min et al.[3].

A promising method for studying the stability and folding of membrane proteins that addresses many of the difficulties that arise as a result of the presence of the bilayer is single-molecule force spectroscopy on transmembrane proteins embedded in bicelles[3–5]. The first published application of single-molecule force spectroscopy to a membrane protein in bicelles focused on the $\alpha$-helical intramembrane protease GlpG[3]. Min et al. found that GlpG unfolds cooperatively at high force and also refolds reliably at low force. During unfolding at high force, Min et al. sometimes observed transient stalling at intermediate states after GlpG overcame the main barrier to unfolding. The changes of the end-to-end distance during these subglobal unfolding events correspond to the size of helical hairpins. By analyzing the effect of mutations on the probability of observing these intermediate states, they determined that the unfolding of GlpG at high force proceeds from the C-terminus to the N-terminus. Although they were unable to observe unfolding and refolding at a single value of the applied force, which would have allowed for a direct determination of the stability at that force, by extrapolating the unfolding and refolding data (obtained in separate force regimes) to zero force, they were able to reconstruct a putative zero-force free energy profile along the end-to-end distance. The inferred landscape indicates that GlpG has a high-kinetic stability $\left(\Delta G_u^\dagger = 21.30\,\mathrm{kT}\right)$ and a relatively low-thermodynamic stability $(\Delta G = 6.54\,\mathrm{kT})$. They also obtained distances to the transition state from the folded state $(\Delta x_f^\dagger = 14.8\,\text{Å})$ and from the unfolded state $(\Delta x_u^\dagger = 35.6\,\text{Å})$, which together indicate that the end-to-end distance change during the rate-limiting step of refolding is 50.4 Å. Although Min et al. did not identify a specific structural transition that they thought gave rise to the observed end-to-end distance during the rate-limiting step of refolding, they argued that refolding occurs as a single cooperative step wholly within the bilayer and that this high degree of cooperativity may be an evolved safeguard against the pathological effects of populating misfolded or partially folded states.

In this study, we use a structure-based protein folding forcefield encoding a well-funneled landscape[6] along with an implicit bilayer potential to model the energy landscape of GlpG in the presence of a bilayer (Supplementary Figure 1 and Supplementary Note 1). This same type of forcefield has explained GlpG's puzzling negative $\phi$-values[2,7] measured experimentally[8] that turn out to involve backtracking[9], i.e., breaking up native interactions in order to complete folding[10]. The implicit bilayer model used in the present study is more elaborate than the model that was used in the previous study of GlpG's negative $\phi$-values (see the Methods section for details). To infer folding and unfolding mechanisms and to visualize the free energy landscape both in the low-applied force and high-applied force regimes, we plotted two-dimensional free energy profiles as a function of the end-to-end distance, $D$, and the average $z$-value of the $C_\alpha$ atoms, $Z$. The folded state is found to have high-kinetic stability and low-thermodynamic stability and the end-to-end distance change during the rate-limiting step of refolding on the computed landscape agrees quantitatively with the end-to-end distance change inferred from the unfolding and refolding experiments. We also find a C-terminal-first unfolding mechanism that proceeds in a highly cooperative manner at high force in steps corresponding to the extraction of helical hairpins from the bilayer, as was inferred by experiment. The refolding mechanism implied by the computed free energy landscapes, however, is quite different from that which was presumed in the original force spectroscopy study. This reinterpretation of the refolding mechanism, if confirmed by further experimental studies, would substantially change the meaning of the measured kinetic and thermodynamic stabilities. The good agreement between the existing experiments and our calculations suggests that the rate-limiting step for refolding is the simultaneous insertion and folding of transmembrane helices 5 and 6 starting from a state with helices TM1–4 inserted and folded. This structural mechanism of the rate-limiting step implies that the thermodynamic stability inferred by the force spectroscopy experiments may correspond to the stability of the fully folded state relative to a partially inserted metastable state with transmembrane helices 5 and 6 remaining on the bilayer interface. These results highlight the highly nontrivial nature of measuring transmembrane protein stability in bilayer environments. We find no evidence that a fully and correctly inserted but still unfolded state is populated to any significant degree at low force as is envisioned in the two-stage folding hypothesis. The stability of the fully folded state relative to an otherwise unfolded state with complete and correct topological insertion of the transmembrane helices may be significantly higher than the free energy difference between the folded state and the partially inserted metastable state that was measured by force spectroscopy.

## Results

**A metastable nonnative state is populated during refolding.** Figure 1 shows the free energy landscape of GlpG in the presence of a bilayer as a function of $Z$, the average of the $z$-coordinates of the $C_\alpha$ atoms, and $D$, the distance between the two termini of GlpG (hereafter referred to as the end-to-end distance). Conformations with all transmembrane helices inserted in the membrane have $Z$ values between 0 and $-5$ Å. As transmembrane helices are pulled out of the membrane, $Z$ becomes more and more negative. $D$ is directly comparable to the end-to-end distances measured by Min et al. Fully inserted and folded conformations of GlpG have $D$ values between 30 and 40 Å. As the transmembrane helices are pulled apart, $D$ increases, with fully extended conformations having $D$ values of $\approx$300 Å.

We see in Fig. 1 that the folded ensemble (N) has a $Z \approx -3$ Å and a $D \approx 35$ Å. At low-applied force, the folded state is the global

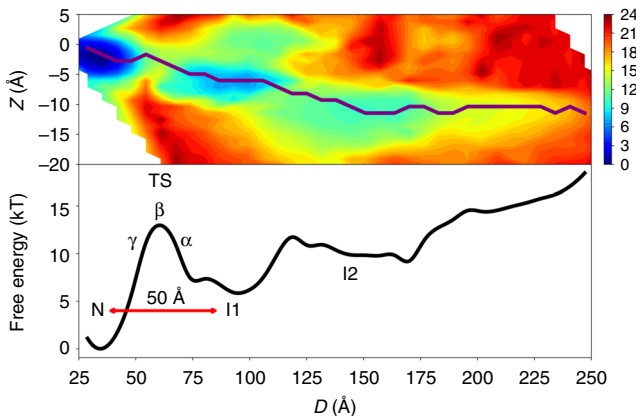

**Fig. 1** Free energy landscape underlying GlpG's folding and insertion into a bilayer under low-applied force (Top) Two-dimensional free energy landscape at low-applied force as a function of $Z$, the average of the $z$-coordinates of the $C_\alpha$ atoms, and $D$, the distance between the N-terminus and C-terminus of GlpG. Relative free energies are indicated with colors in units of kT, where blue indicates a low free energy and red indicates a high-free energy. A folding path is shown as a purple line drawn from a highly extended state ($Z \approx -10$ Å, $D \approx 250$ Å) to the folded state ($Z \approx -3$ Å, $D \approx 37$ Å). Two metastable ($I1$ and $I2$) states are present at intermediate values of $D$ and values of $Z$ that are more negative than the average $Z$ of the folded state. (Bottom) One-dimensional free energy profile along the path shown in the top panel as a function of $D$. A red double-headed arrow that is 50 Å in length is shown between $N$ and $I1$. This distance corresponds to the end-to-end distance change during the rate-limiting step of refolding at low force. The first metastable state, $I1$, is about 6.5 kT less stable than the folded state. $I2$ is about 9 kT less stable than the folded state. Three locations ($\alpha$, $\beta$, and $\gamma$) on the path near the transition state (TS) are also indicated. Representative structures from $I1$, $\alpha$, $\beta$, and $\gamma$ are shown in Fig. 2. Representative structures for $I2$ are shown in Fig. 4

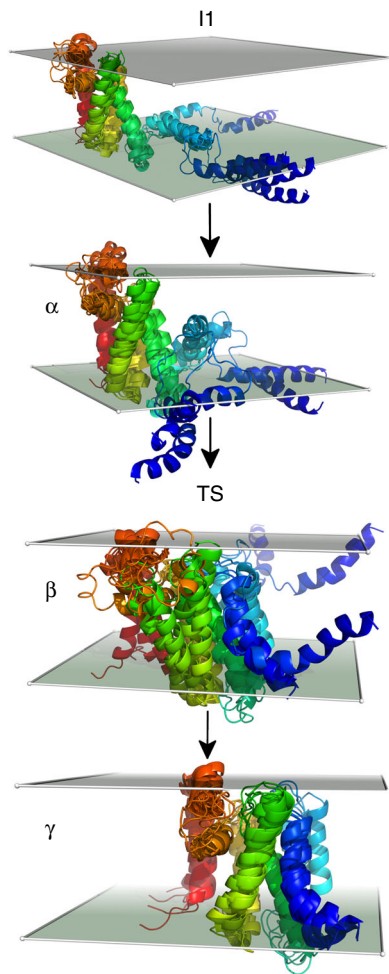

**Fig. 2** Structural mechanism of the rate-limiting step of refolding at low force. Progression of folding is shown from top to bottom. The structure labels ($I1$, $\alpha$, $\beta$, $\gamma$, and TS) are the same as those used in Fig. 1. The structure of GlpG is colored according to sequence index from red (N-terminal, TM1) to blue (C-terminal, TM6). TM1 is red, TM2 is yellow, TM3 is yellow-green, TM4 is green, TM5 is light blue, and TM6 is dark blue. For each state, several representative structures are aligned and overlayed. Translucent panels are shown to indicate the locations of the upper and lower bilayer interfaces. In $I1$, TM5–6 are on the bilayer interface. As folding proceeds, TM5 and then TM6 are pulled into the bilayer and fold onto TM1–4

free energy minimum. A metastable state with $Z \approx -6$ Å and $D \approx 87$ Å is also relatively low in free energy (about 6.5 kT higher than the folded ensemble). This metastable state, which we call $I1$ for consistency with the notation in Min et al.[3], is separated from the folded ensemble by a high barrier of approximately 13 kT. Structures sampled from this metastable state have TM1–4 inserted into the membrane and TM5–6 extracted from the membrane. Higher still in free energy is $I2$, an ensemble having TM1–2 inserted and TM3–6 extended on the bilayer interface.

**The rate-limiting step of refolding is insertion of TM5–6.** In order to analyze the structural mechanism of folding, we first identified a plausible, low-free energy folding pathway from a highly extended state to the folded state (Fig. 1, top panel). Details of how the path was determined can be found in the Methods section. At low force, folding proceeds largely downhill from highly extended states until $I1$ is reached ($D \approx 87$ Å, Fig. 1, bottom panel). At this point, a relatively large free energy barrier ($\approx 7$ kT) must be overcome. The rate-limiting step of folding involves pulling TM5–6 into the membrane and folding this helical hairpin onto TM1–4 (Fig. 2). On either side of the barrier (at positions $\alpha$ and $\gamma$ along the folding path as shown in Figs. 1, 2), TM1–4 are folded. The structures selected from around the barrier peak (position $\beta$) show more variability than those on either side of the barrier peak, indicating that GlpG must partially unfold to bring TM5–6 in from the bilayer interface. Analyses of the expectation value of the $V_{AMH-Go}$ energy term along the folding pathway (Supplementary Figure 2 and Supplementary Note 2), the average values of structural order parameters

computed for ensembles along the folding pathway (Supplementary Table 1), and average contact maps computed for ensembles along the folding pathway (Supplementary Figure 3 and Supplementary Note 3) all indicate that GlpG must partially unfold when transitioning from $\gamma$ to $\beta$. This behavior is reminiscent of the backtracking seen in the detergent micelle-mediated folding of GlpG that leads to a large number of negative $\phi$-values[8,9]. This final folding transition changes the end-to-end distance from $D \approx 87$ Å to $D \approx 37$ Å, corresponding to a change in the end-to-end distance of approximately 50 Å.

**Unfolding by extraction of helical hairpins from the bilayer.** The free energy landscape of GlpG under high applied force is shown in Fig. 3. At high-applied force, $I1$ is only weakly metastable. Once the barrier between the $N$ and $I1$ has been overcome, unfolding proceeds largely downhill from $I1$ and through $I2$ to $U$, a highly extended state. The intermediate states $I1$ and $I2$ are related to the folded state $N$ by successive extraction of helical

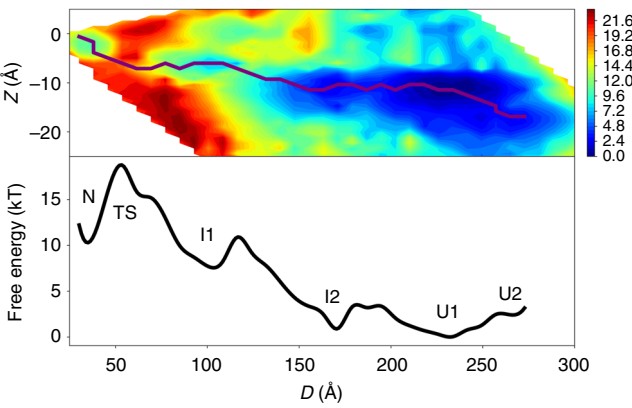

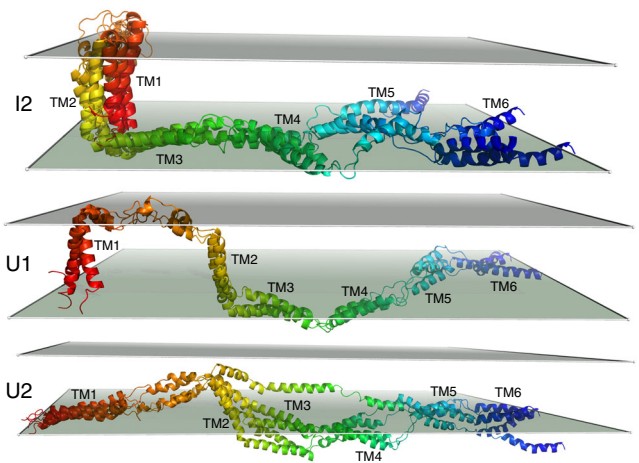

**Fig. 3** Free-energy landscape underlying GlpG's unfolding and extraction from a bilayer under high applied force (Top) Two-dimensional free energy landscape at high applied force as a function of $Z$, the average of the $z$-coordinates of the $C_\alpha$ atoms, and $D$, the distance between the N-terminus and C-terminus of GlpG. Relative free energies are indicated with colors in units of kT, where blue indicates a low-free energy and red indicates a high-free energy. An unfolding path is shown as a purple line drawn from the folded state, N, ($Z \approx -3$ Å, $D \approx 37$ Å) to the unfolded state, U ($Z \approx -17$ Å, $D \approx 270$ Å). (Bottom) One-dimensional free energy profile along the path shown in the top panel as a function of the end-to-end distance, $D$. Representative structures from TS and $I1$ are shown in Fig. 2. Representative structures from $I2$, $U1$, and $U2$ are shown in Fig. 4. Under these high force conditions, the folded state is metastable and is separated by a large barrier from the first intermediate state, $I1$. The barriers separating the intermediate states $I1$, $I2$, and $U$ are significantly smaller than the barrier between $N$ and $I1$. At this particular value of the applied force, the completely extended state $U2$ is slightly higher in free energy than $U1$, which has TM1–2 inserted in the membrane but unfolded (see Fig. 4). At larger values of the applied force, $U2$ becomes the global free energy minimum

**Fig. 4** Representative structures of GlpG at high values of $D$, the end-to-end distance. The structure labels ($I2$, $U1$, and $U2$) are the same as those used in Fig. 3. The structure of GlpG is colored according to sequence index from red (N-terminal, TM1) to blue (C-terminal, TM6). TM1 is red, TM2 is yellow, TM3 is yellow-green, TM4 is green, TM5 is light blue, and TM6 is dark blue. The helices are also labeled with text. For each state, several representative structures are aligned and overlayed. Translucent panels are shown to indicate the locations of the upper and lower bilayer interfaces. In $I2$, TM3–6 are on the bilayer interface and TM1–2 are inserted and folded. In $U1$, TM1-2 are unfolded but still inserted in the bilayer. In $U2$, none of the helices are fully inserted into the bilayer. For clarity, all of the structures have been aligned, but only a single location of the upper and lower bilayer interfaces are shown. Therefore, particularly for structures that are difficult to align such as those in $U2$, the locations of the bilayer interfaces relative to the structures should be considered approximate

hairpins from the membrane bilayer starting from the C-terminus (see Figs. 2, 4). Only under these high-force conditions do we see a significant population of states that are both extended and partially inserted. Example of such structures at ($Z \approx -3$ Å, $D \approx 220$ Å) are shown in Supplementary Figure 4 (see also Supplementary Note 4). Instead of the transmembrane helices being pulled apart while adopting native-like orientations with respect to the membrane, the commonly presumed starting point for the second stage of membrane protein folding, these states have a nonnative arrangement of transmembrane helices with respect to the membrane. This ensemble of structures with incorrect topology with respect to the membrane is less favorable than the ensemble $U1$, which has the two most hydrophobic transmembrane helices in GlpG, TM1 and 2, embedded in the membrane. In contrast, the ensemble at ($Z \approx -3$ Å, $D \approx 220$ Å) has the second and third most hydrophobic helices, TM2 and 5, embedded in the membrane at the same time that the other helices are segregated to opposite membrane surfaces. Thus, we see that establishment of the correct orientation of the helices in the membrane is favored over nonnative alternatives even at the earliest stages of folding and insertion.

## Discussion

The barriers for transitions between $N$ and $I1$ in our computed landscapes are somewhat smaller than those that were measured in the force spectroscopy experiments of Min et al. For example, under conditions where $N$ is 6.5 kT more stable than $I1$, the barrier to unfolding was measured to be 21 kT by force spectroscopy, whereas

the corresponding value in our computed landscape is 13 kT. Energetic quantities are expected to be less robust with respect to the parameterization of the simulation model than are the structural geometric quantities associated with folding transitions, such as the contact probabilities in transition states (i.e., $\phi$-values) and end-to-end distance changes upon insertion and folding of helical hairpins. Although, we currently lack the detailed biophysical studies that would be necessary uniquely to determine optimal model parameters for the implicit membrane model, it is notable that we were unable to find any set of forcefield parameters that gave a free energy landscape that would be consistent with the two-stage picture of membrane protein folding. While developing the current implicit membrane model, all preliminary tests that we performed that involved increasing or decreasing the strength of individual energy terms by between 10 and 100% compared to the values given in the Methods section failed to produce a landscape having folded and unfolded states that were completely inserted into the membrane but that were nevertheless approximately equal in free energy and also separated by a significant barrier. The structural details of the folding and unfolding mechanisms shown in Figs. 1–4 appear to be largely determined both by the native structure of GlpG and by the presence of a bilayer. In the absence of more detailed measurements and careful calibration of simulation model parameters, obtaining quantitatively accurate barrier heights in the presence of bilayers, however, will apparently require a combination of experimentation and theory. Analyses both of the expectation values of the energy terms in the current forcefield and of the effects of perturbing the strengths of these terms on the free energy landscape are given in Supplementary Figures 2, 5 (see also Supplementary Note 2).

Due to the high-kinetic stability of GlpG in a bicelle, in order to observe the unfolding of folded GlpG on reasonable

timescales in the laboratory, the free energy of the transition state must be lowered significantly relative to that of the folded state. Because the transition state has an end-to-end distance that is close to that of the folded state (within 15 Å), a relatively large force must be applied. Applying this force has the effect of tilting the entire landscape and heavily favoring highly extended states in free energetic terms. We note that a back-of-the-envelope calculation suggests that a change in the applied force of 5 pN (the difference between the two force regimes used to measure folding and unfolding rates in the force spectroscopy experiments of Min et al.) is expected to change the relative free energy by 30 kT of two states that differ in extension by 250 Å (such as I1 and U) at 300 K. The largest value of the force that was used to measure refolding rates in the study by Min et al. was 7 pN. The lowest value of the force that was used to measure unfolding rates was 12 pN. We see on the basis of our back-of-the-envelope calculation that this seemingly small gap between the two force regimes that were employed gives rise to large changes in the relative free energy of near native states versus highly extended states. In particular, the highly extended states are expected to be quite high in free energy throughout the force range that was used to measure refolding rates, consistent with what we see in our computed free energy profiles. These considerations alone suggest that the unfolded state reached at high applied force would likely not be the same as the starting point for refolding at low force. The fact that the force spectroscopy measurements indicate that there is a large difference in the changes of the end-to-end distance during high force unfolding (>200 Å) and low force refolding (≈50 Å) means that there are structural differences between the unfolded state that is favored at high force and the unfolded state that is favored at low force. This fact raises the question, "Is the unfolded state favored at low force simply a more generically compact version of the unfolded state favored at high force, or can a specific partially folded structure account for this difference?" The answer to this question has important implications for the interpretation of the rates measured and stability inferred by force spectroscopy. In the current study, we have identified I1, the state with TM5–6 on the bilayer interface, as the starting point for refolding at low force. Our simulation results suggest that the thermodynamic stability inferred by experiment is, therefore, not the relative stability of the folded and completely unfolded states but, instead, reflects the relative stability of N and I1. According to the steric trapping experiments in detergent micelles[11], the C-terminal half of GlpG indeed has a lower stability than the N-terminal half. The stability measured by force spectroscopy in bicelles (≈6.5 kT) is closer to the stability measured by steric trapping for the C-terminal half of GlpG (TM4–6, $\Delta G \approx 8.0$ kT) than for the N-terminal half (TM1–3, $\Delta G \approx 9.8$ kT). The stability of N with respect to U may be significantly higher than was initially inferred by force spectroscopy. The high-kinetic barrier separating N and I1, the origin of which was unclear within the two-stage picture of membrane protein folding, is now seen to be associated with the simultaneous insertion and folding of TM5–6 from the bilayer interface in the folding direction and the unfolding and extraction from the bilayer in the unfolding direction. GlpG has, in recent years, become a heavily studied model system of transmembrane protein folding and stability[3,8,9,11–14]. These studies have provided unprecedented detail regarding the folding of a transmembrane protein in a variety of conditions. Nonetheless, measuring membrane protein stability and determining detailed folding mechanisms in the presence of bilayers have remained major challenges. The results of the current study highlight the highly nontrivial nature of measuring membrane protein stability and folding

mechanisms in bilayer-like environments. The broken translational and rotational symmetries and high-kinetic barriers induced by the presence of the bilayer support a multitude of potentially metastable partially folded states that reduce the cooperativity of folding and complicate the very definition of stability. In this case, only by reanalyzing sensitive single-molecule experiments in light of detailed simulations were we able to arrive at a satisfactory structural explanation for the striking character of the energy landscape inferred by experiment.

The importance of the helical hairpin as a unit of membrane protein structures and folding mechanisms was posited by Engelman and Steitz[15], 4 years before the first three dimensional structure of a transmembrane protein was solved[16]. In 1999, Booth and Curran[17], when thinking specifically about the case of in vitro spontaneous refolding of Bacteriorhodopsin, suggested two possibilities for the rate-limiting step of refolding. One of the mechanisms that Booth and Curran put forward in 1999 involves pre-formation of the N-terminal part of Bacteriorhodopsin and a rate-limiting step of cooperative insertion and folding of the two C-terminal helices as a helical hairpin, exactly as we see here for GlpG. Unfortunately, because Bacteriorhodopsin has an odd number of transmembrane helices and, therefore, has its two termini on opposite sides of the membrane in the folded structure, the force spectroscopy experiments performed recently on GlpG are not straightforwardly possible on Bacteriorhodopsin. To our knowledge, in the subsequent two decades since Booth and Curran published their speculations, it has not been resolved whether or not insertion of the C-terminal hairpin is rate-limiting for Bacteriorhodopsin during in vitro refolding. Booth and coworkers have recently probed co-translational folding of GlpG into membranes in the absence of the translocon using infrared spectroscopy[13], a situation that is somewhat analogous to the situation explored in the force spectroscopy refolding experiments of Min et al. The infrared spectroscopy experiments suggest that, while GlpG is being translated, helices form, these helices insert into the membrane, and some tertiary structure forms. The infrared spectroscopy measurements themselves cannot differentiate between helix formation signals coming from different transmembrane helices, but Booth and coworkers suggest, based on the observation that helices TM1–2 of GlpG are significantly more hydrophobic than other pairs of helices in GlpG, that the first transmembrane helices to insert into the membrane are likely to be TM1–2. These observations and inferences by Booth and coworkers are consistent with the folding mechanism described above in the Results section.

The translocon has for some time been thought of as a protein conducting channel that co-translationally guides newly synthesized hydrophobic polypeptides into or across the membrane. The most commonly presumed structural mechanism is that hydrophobic transmembrane helices are inserted into the translocon channel and exit through a dynamic lateral gate. In light of the refolding mechanism found for GlpG in this work, it is interesting to consider the new view of translocon-mediated insertion put forward by Cymer et al.[18]. This new view starts from the well-established facts that newly synthesized transmembrane helices have a high affinity for and close proximity to the membrane interface and goes on to suggest that, instead of transmembrane helices entering the translocon pore and later being ejected from the lateral gate, the translocon could serve primarily as a catalyst that facilitates the membrane crossing of polar loops that connect pairs of transmembrane helices as the transmembrane helices slide along the outside of lateral gate without ever actually entering the translocon pore. This sliding mechanism has shown up in

theoretical models of translocon-assisted transmembrane helix insertion that are in harmony with experimental observations of integral membrane protein topology[19–21]. When the energy cost of inserting a helical hairpin into the membrane is not sufficiently offset by the energy gain of insertion of a highly hydrophobic helical hairpin and formation of strong native contacts (as is the case for TM5–6 in GlpG, which are only modestly hydrophobic and have relatively few native contacts with each other in the folded structure of GlpG), then a large barrier results and spontaneous insertion and folding becomes slow. Our current results suggest that refolding of GlpG would proceed reliably and rapidly from a bilayer interface-associated state to the fully folded state in the presence of a catalyst for inserting helical hairpins, a role that could be filled in vivo by the translocon without the need for transmembrane helices to ever directly enter the translocon channel, as suggested by Cymer et al. Whether or not a catalyst would be required to ensure proper folding of GlpG in vivo is a quantitative question having to do with the barrier height that limits refolding (≈15 kT according to the force spectroscopy experiments) and the rates of other processes taking place in the cell, such as aggregation and degradation, that might interfere with the completion of folding.

The two-stage model of transmembrane protein folding, wherein membrane protein folding occurs, at least conceptually, in two distinct stages of insertion and helix packing, was put forward by Popot and Engelman[22]. While very useful as a thermodynamic and conceptual model of the way membrane proteins might fold, the validity of the two-stage model as a general kinetic description of the way membrane proteins fold in the presence of bilayers has been largely untested. The present simulations along with their harmony with experiment suggest that folding and insertion are coupled at every step of GlpG folding into a bilayer. Determining whether or not the in vivo folding mechanism of GlpG is essentially a translocon-catalyzed version of the in vitro mechanism or, instead, follows a two-stage model in which insertion entirely precedes folding will require further experiments and simulations.

The idea that the lowest free energy nonnative state of a transmembrane protein in a bilayer is a partially inserted state has significant implications for our understanding of transmembrane protein evolution, degradation, and design. The study of membrane protein quality control and degradation is only in its infancy, but it is already becoming clear that transmembrane helix hydrophobicity, independent of folded structure stability, is an important factor in determining the degradation rate due to the fact that soluble proteases must dislocate the transmembrane helices from the membrane in order to perform the degradation[14]. In order to avoid becoming unfolded and risking degradation, membrane protein sequences must therefore evolve to both fold and remain stably inserted in the membrane after being inserted into the membrane by the translocon. Structural knowledge of the lowest free energy nonnative states can be useful for informing protein design algorithms. The emerging picture of metastable partially inserted states of membrane proteins may prove useful for informing membrane protein design algorithms through the explicit negative design against such states.

Since the publication of the force spectroscopy experiments on GlpG[3], the same method has been applied to a designed α-helical transmembrane protein with four transmembrane helices[5] and a large chloride transporter, ClC, that has a complex topology and two independently metastable subdomains[4]. In the case of the designed transmembrane protein, unfolding occurs cooperatively in a single step and refolding after force-induced unfolding was found to be reliable and to occur in two steps. By summing the stabilities inferred via measuring the folding and unfolding rates for both steps, a total stability of ≈13 kT was determined. The simple and symmetric topology involving the lateral association of two helical hairpins provides a plausible structural rationale for the observation of two step refolding. Whether or not the intermediate and unfolded structures involve extraction of helical hairpins from the membrane or simply dissociation of the helices in this case is not known. A comparison given in ref. [5] of the stability per helix between the designed transmembrane protein (≈3.4 kT per helix), GlpG (≈1.4 kT per helix), and Bacteriorhodopsin (≈2.9 kT per helix, measured by steric trapping[23]) suggests that GlpG has a significantly lower stability per helix than the other two proteins do. In light of the results in this study, however, if one distributes GlpG's apparent stability over just two helices (TM5 and 6) instead of six transmembrane helices, the stability per helix of GlpG rises to ≈2.7 kT, which is approximately equal to the stability per helix that was reported for Bacteriorhodopsin based on steric trapping experiments. Force-induced unfolding of ClC[4] occurs in three steps, the first of which is the reversible dissociation of the two subdomains. The subsequent two steps correspond to unfolding of the two subdomains. This mechanism is consistent with the proposed evolutionary history of ClC, which is thought to involve the evolution of two independently stable domains that were subsequently fused together. When attempting refolding from the force-induced unfolded state, it was found that only a small fraction (≈11%) of refolding attempts lead to successful and complete refolding. When compared to the reliable refolding found for GlpG[3] and the designed transmembrane protein[5], these latest results on ClC suggest that transmembrane proteins with more complex topologies become deeply trapped in topologically nonnative states upon spontaneous refolding from the interface. Without being able to measure refolding rates, the stability of ClC could not be determined. During refolding, including in refolding attempts that apparently result in nonnative topologies, ≈100 Å compactions were observed. The cooperative insertion of multiple transmembrane helices seems a likely explanation for these observations, but confirmation of this hypothesis and a determination of the nonnative topologies formed during refolding await more detailed experimental and theoretical studies.

## Methods

**Combined protein-implicit bilayer forcefield.** The structure-based forcefield used in this study is a variant of the forcefield used to study soluble proteins[24] that has been modified for use with membrane proteins[9]. The same structure-based forcefield used in the present study was previously used in a prior study to elucidate the origin of the puzzling preponderance of GlpG's negative $\phi$-values[2,7] when GlpG is folded in mixed detergent micelles[8,9]. Refolding in micelles turns out to involve backtracking[10], which gives rise to negative $\phi$-values. $\phi$-values are measured experimentally by comparing the change in the apparent stability of the transition state (by measuring changes in folding rates) to the change in the stability of the folded state upon making a mutation. GlpG's negative $\phi$-values arise from mutations that both decrease the stability of the folded state and increase the folding rate by allowing more facile backtracking. In ref. [9], it was shown that the large number of negative $\phi$-values found for GlpG folding in micelles could be attributed to a multistep folding mechanism that involves breaking and eventually reforming an interface while proceeding in the folding direction (backtracking). This folding complexity was also shown to be partially attributable to GlpG's modular structure and also to the high degree of conformational entropy in the micellar unfolded state.

The implicit bilayer potential used in the present study is substantially more elaborate than the implicit bilayer potential used previously for membrane protein structure prediction[25,26] and in the analysis of the micelle-mediated GlpG folding[9]. In addition to containing a residue type-dependent membrane burial term, the current version of the implicit bilayer potential includes a term orienting each helix and a lipid-mediated interaction between pairs of helices[27]. The orientation term favors alignment of each transmembrane helix with the membrane normal, since such conformations only minimally disrupt to the lipid

bilayer's natural liquid crystalline ordering. Transmembrane helices, as inclusions in the membrane, induce local fluctuations in the density of lipids in the surrounding bilayer. Interaction of these density fluctuations induced by two helices leads to a pairwise nonmonotonic effective interaction between the helices[27]. For DMPC, the lipid used to form bicelles in the experiments of Min et al., this interaction is repulsive at pair-distances between 10 and 25 Å and is attractive at shorter distances for inclusions that are the size of transmembrane helices. All of the terms in the implicit bilayer potential switch off smoothly when going from the transmembrane region to the extramembrane region, which is important for modeling folding and unfolding events that are coupled to insertion and extraction of transmembrane helices, a key aspect of the current study.

The total potential energy function used to simulate the GlpG-bilayer system is given in Eqs. (1), (2).

$$V_{total} = V_{SBM} + V_{bilayer}. \tag{1}$$

$$V_{bilayer} = V_{burial} + V_{orientation} + V_{helix-pair}. \tag{2}$$

In Eq. (1), $V_{SBM}$ is a structure-based model describing the direct interactions of the protein chain with itself and $V_{bilayer}$ is an implicit bilayer potential that describes the influence of the bilayer on the protein. Each term is described in the following sections and in the manuscripts we reference.

**Structure-based protein model.** The structure-based model used to describe the direct interactions of the protein chain with itself is based on a model that was previously used to study the folding mechanisms of soluble proteins[24] and that was subsequently modified for use with α-helical transmembrane proteins[9]. Eq. (3) shows the different terms in $V_{SBM}$.

$$V_{SBM} = V_{con} + V_{chain} + V_{\chi} + V_{rama} + V_{excl} + V_{AMH-Go}. \tag{3}$$

In Eq. (3), the backbone terms $V_{con}$, $V_{chain}$, $V_{\chi}$, $V_{rama}$, and $V_{excl}$ are responsible for ensuring that the backbone adopts protein-like conformations and does not overlap with itself. These potentials are described in detail in the Supporting Information of ref. [28]. GlpG's native secondary structure was determined using the STRIDE algorithm[29,30] and was used as input to the Ramachandran dihedral angle term, $V_{rama}$, to provide an additional bias as described in ref. [28]. The functional form of $V_{AMH-Go}$ is given in Eqs. (4)–(9) and in the manuscript that first described the model[24].

$$V_{AMH-Go} = -\frac{1}{2} \sum_i |E_i|^p. \tag{4}$$

$$E_i = \sum_j \varepsilon_{ij}(r_{ij}). \tag{5}$$

$$\varepsilon_{ij}(r_{ij}) = -\left| \frac{\varepsilon}{a} \right|^{1/p} \Theta\left(r_c - r_{ij}^N\right) \gamma_{ij} \exp\left[ -\frac{\left(r_{ij} - r_{ij}^N\right)^2}{2\sigma_{ij}^2} \right]. \tag{6}$$

$$a = \frac{1}{8N} \sum_i \left| \sum_j \gamma_{ij} \Theta\left(r_c - r_{ij}^N\right) \right|^p. \tag{7}$$

$$\sigma_{ij} = |i - j|^{0.15} \text{ Å}. \tag{8}$$

$$\gamma_{ij} = \begin{cases} \gamma^{short}, & \text{if} |i - j| < 5. \\ \gamma^{long}, & \text{otherwise.} \end{cases} \tag{9}$$

In Eq. (4), the sum runs over all $C_\alpha$ and $C_\beta$ atoms $i$, and $E_i$ is the energy of atom $i$. In Eqs. (4), (6), (7), $p$ is a nonadditivity exponent. In this study and in the previous analysis of GlpG folding[9], a value of $p = 1$ was used, resulting in a pairwise additive model. In Eqs. (5), (6), $r_{ij}$ is the distance between the atoms $i$ and $j$ and $r_{ij}^N$ is the corresponding distance in the native structure. $\varepsilon$ is a scaling factor. In this study a value of $\varepsilon = 0.8$ kcal/mol was used. $\Theta$ is the Heaviside function, and a cutoff of $r_c = 7$ Å between atoms was used to define native contacts. $\sigma_{ij}$ is a sequence separation-dependent interaction well width, whose precise form is shown in Eq. (8). $a$ is a normalization constant, and $N$ is the total number of residues ($N = 181$ for GlpG). $\gamma_{ij}$ are interaction weights that depend on the sequence separation, $|i - j|$. For this study and for the previous study of GlpG folding[9], we have set $\gamma^{short} = 1.0$ and $\gamma^{long} = 0.5$ such that the local-in-sequence (helical) contacts are strengthened relative to the nonlocal-in-sequence contacts. The practical effect of this tuning of the model is that helices are very stable and tend not to break even

when the tertiary structure of the protein unfolds, which is an appropriate description of GlpG's transmembrane helices when they are embedded in detergent micelles[8] and, presumably, also when they are embedded in lipid bilayers without very strong external forces being imposed. At forces exceeding 15pN, Min et al. did observe a helix-to-coil transition[3] after unfolding. According to the model of GlpG's folding and unfolding mechanisms described here, the helices would be on the interface and partially exposed to solvent after unfolding. The helices in the current simulation model do not break even at forces sufficient to globally favor the fully unfolded state (see Fig. 4 in the main text). This is probably due, at least in part, to the fact that, unlike the other terms in the protein-bilayer described below, the strengths of the interactions in $V_{AMH-Go}$ are not taken to be z-dependent. One would expect that, due to the larger number of potential hydrogen bonding partners available in the aqueous phase, helices would be easier to unravel once they had been extracted from the bilayer, but the current model does not take this effect into account. As a result, the values of $D$ for the intermediate and unfolded states in Fig. 3 of the main text are all somewhat lower than the changes in the end-to-end distances measured during the unfolding transitions at 21pN by Min et al.[3] would suggest. This feature of the model is not expected to significantly influence the refolding mechanism at low force because refolding only occurs at a significant rate when the applied force is below the force at which the helix-coil transition would be seen.

**Structure and sequence of the GlpG construct.** The native distances used as input for the structure-based model were taken from a structure of GlpG's transmembrane domain (residues 91–271) and, for the sequence-dependent potential energy terms, the sequence was the wild-type sequence taken from the same structure (PDB ID: 2XOV).

**Transmembrane helices of GlpG.** There are six transmembrane helices in GlpG (TM1–6). The residue ranges used in the current study are the same as those denoted in the force spectroscopy study of Min et al.[3]: TM1: 94–114, TM2: 147–168, TM3: 171–192, TM4: 200–217, TM5: 226–241, TM6: 250–269. GlpG's two interfacial helices in the large loop, L1, are not considered transmembrane helices for the purposes of $V_{orientation}$ and $V_{helix-pair}$, though, like all residues, the residues in L1 do have a sequence-dependent membrane burial preference through $V_{burial}$.

**Perturbation to uniform AMH–Go interaction strengths.** Upon initial examination of the free energy profiles as a function of $D$ and $Z$, the inferred folding path indicated that folding and insertion would proceed downhill at low values of the applied force, which would be in obvious contradiction to the force spectroscopy measurements. Structural analyses of the basins along the low-free energy folding pathway revealed, however, that the apparent lack of a barrier was due to the existence of several near-native ensembles overlapping the transition state in their $(D, Z)$ values. Details of these near-native ensembles and an analysis of the free energy landscape using an alternative set of order parameters can be found in the Supplementary Information (Supplementary Figures 6 and 7 as well as Supplementary Notes 5 and 6). Using perturbation theory to preferentially enhance the stability of the N-terminal part of GlpG by 20% helped to clarify the folding pathway in $(D,Z)$ space. We emphasize that the choice of a 20% perturbation is not arbitrary but is a value quantitatively consistent with measurements of subglobal stabilities of GlpG by steric trapping, which indicate that the N-terminal half of GlpG is more stable than the C-terminal half[11]. The resulting energy landscape that includes the 20% perturbation recapitulates all of the major observations of the force spectroscopy study.

The molecular dynamics simulations were performed using a model with uniform contact interaction strengths for $V_{SBM}$, with the only variation arising from the sequence separation dependence. It is important to note, however, that in computing the free energy landscapes shown in Figs. 1, 3, the strengths of interactions within the N-terminal 4 helices of GlpG (TM1–4, residues 91–217) were increased by 20% over the uniform values and perturbation theory was used to compute the profiles (see Section 5 for a discussion of the free energy calculations using the multistate Bennett acceptance ratio (MBAR) method). This introduction of nonuniformity was motivated by the known fact that the N-terminal half of GlpG is more stable than the C-terminal half[11]. The uniform model itself generates several ensembles of structures (see Supplementary Figure 6 and Supplementary Note 5) that overlap the transition state in $(D,Z)$ coordinate space but that are not structurally intermediate between $I1$ and $N$. These partially folded states involve separating TM1 from TM2–6 or separating TM1–3 from TM4–6. Increasing the strength of the interactions in TM1–4 by 20% has the effect of raising the apparent height of the barrier along the folding path in $(D,Z)$ space by disfavoring these partially folded states that are actually in the native kinetic basin but overlap the $I1 \rightarrow N$ transition state in $(D,Z)$ coordinate space. Whether or not the apparent stability of these partially folded near-native states is an incorrect result arising from the assumption that native contacts have equal weights or whether, in fact, they actually are populated in experiment and could be detected by more sensitive experiments at very low applied force remains an open question.

**Table 1 Amino acid hydrophobicity scale used in $V_{burial}$**

| Amino acid type | Ala | Arg | Asn | Asp | Cys | Gln | Glu | Gly | His | Ile |
|---|---|---|---|---|---|---|---|---|---|---|
| $\Delta G_{water-octanol}$ (kcal/mol) | 0.5 | 1.81 | 0.85 | 3.64 | −0.02 | 0.77 | 3.63 | 1.15 | 0.11 | −1.12 |
| Amino acid type | Leu | Lys | Met | Phe | Pro | Ser | Thr | Trp | Tyr | Val |
| $\Delta G_{water-octanol}$ (kcal/mol) | −1.25 | 2.8 | −0.67 | −1.71 | 0.14 | 0.46 | 0.25 | −2.09 | −0.71 | −0.46 |

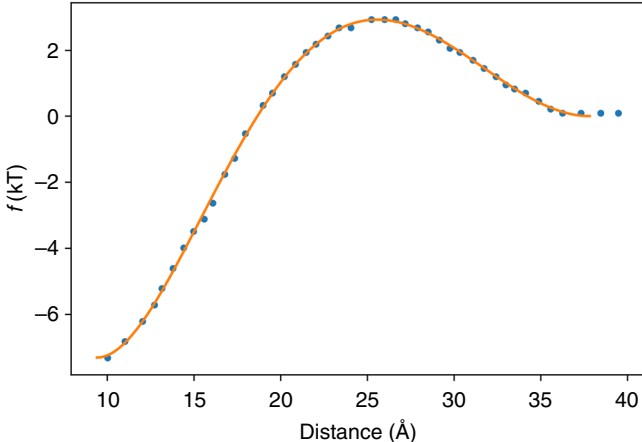

**Fig. 5** Functional form of $f(r_{ij})$ given in Eq. (18). The orange line is a spline fit to the data obtained from ref. [27] for two 5 Å inclusions embedded in a DMPC bilayer

**Switching function for implicit bilayer potentials**. All parts of the implicit bilayer potential switch off smoothly near the interfaces of the intramembrane region ($-15\,\text{Å} < z < 15\,\text{Å}$) and extramembrane regions ($z < -15\,\text{Å}$ or $z > 15\,\text{Å}$). The switching function, $\Theta$, is given in Eq. (10).

$$\Theta(z_i, z_m) = \left\{ \frac{1}{2}\tanh[k_m(z_i + z_m)] + \frac{1}{2}\tanh[k_m(z_m - z_i)] \right\}. \quad (10)$$

In Eq. (10), $i$ is the residue index, $k_m = 0.2\,\text{Å}^{-1}$ is a parameter that controls the distance over which the switching occurs, and $z_i$ is the $z$-coordinate of the $C_\alpha$ atom of the residues participating in the interaction (or $z$-coordinate of the center-of-mass of the $C_\alpha$ atoms within a helix for the helix-pair potential). $z_m$ is the value of $z$ at which $\Theta = 0.5$.

**Single-residue membrane burial term**. To account for the relative preferences of different amino acid types for occupying either the interfacial or the transmembrane regions of the bilayer, we use a single-residue, amino acid type-dependent membrane burial potential (Eq.(11)).

$$V_{burial} = k_{burial} \sum_i A(\sigma_i)\Theta_{burial}(z_i)$$
$$\Theta_{burial}(z_i) = \Theta(z_i, z_m = 15\,\text{Å}) \quad (11)$$

In Eq. (11), $\sigma_i$ is the residue type of residue $i$, $k_{burial} = 1$, and $\Theta_{burial}$ is the switching function given in Eq. (10) with $z_m = 15\,\text{Å}$. The values of $A(\sigma_i)$ are the amino acid hydrophobicities in the octanol scale of Wimley and White[31–34]. The values of $A(\sigma_i)$ used are given in Table 1.

**Single-helix orientation term**. In order to model the influence of the lipid bilayer's liquid crystal ordering on the orientation of transmembrane helices, we apply a cylindrical radius of gyration term, $V_{orientation}$, to each transmembrane helix individually (Eqs. (12)–(17)). The residues included in each of GlpG's six

transmembrane helices are given in Section 4 above.

$$V_{orientation} = k_{orientation} \sum_j \left( R_g^j \right)^2. \quad (12)$$

$$R_g^j = \sqrt{\frac{\sum_i m_i(r_i - r_{cm}^j)^2 \Theta_{orientation}(z_i)}{\sum_i m_i}}. \quad (13)$$

$$\Theta_{orientation}(z_i) = \Theta(z_i, z_m = 12\,\text{Å})$$
$$r_i = \sqrt{x_i^2 + y_i^2} \quad (14)$$

$$r_{cm} = \sqrt{x_{cm}^2 + y_{cm}^2}. \quad (15)$$

$$x_{cm}^j = \frac{\sum_i m_i x_i}{\sum_i m_i}. \quad (16)$$

$$y_{cm}^j = \frac{\sum_i m_i y_i}{\sum_i m_i}. \quad (17)$$

In Eq. (12), $k_{orientation}$ is a scaling factor for adjusting the strength of the orientation term relative to the other terms in the forcefield. In this study, a value of $k_{orientation} = 0.1\,\text{kcal/mol}/\text{Å}^2$ was used. $R_g^j$ is the cylindrical radius of gyration of helix $j$ (Eq. (13)). In Eq. (13), $i$ is a residue index, the sum runs over residues in helix $j$, and $\Theta_{orientation}$ is the switching function given in Eq. (10) with $z_m = 12\,\text{Å}$. For $V_{orientation}$, a value of $z_m = 12\,\text{Å}$ means that helices are allowed to penetrate about 3 Å into the membrane while lying perpendicular to the membrane normal without incurring a large energy penalty. $r_i$ is the radial coordinate of atom $i$ in the $x$–$y$ plane (Eq. (14)) and $r_{cm}^j$ is the radial coordinate of the center of mass of helix $j$ in the $x$–$y$ plane (Eq. (15)). Eqs. (16), (17) are the equations for the $x$ and $y$-coordinates of the center of mass of helix $j$ in terms of the $x$-coordinates ($x_i$), $y$-coordinates ($y_i$), and masses ($m_i$) of the $C_\alpha$ atoms of residue $i$.

**Helix-pair interaction**. The effect of pairwise lipid-mediated interactions between membrane inclusions has been described by Lagüe, Zuckermann, and Roux in ref. [27]. To model this effect, we employ the pairwise helix–helix potential in Eq. (18).

$$V_{helix-pair} = k_{helix-pair} \sum_{(i,j)} f(r_{ij})\Theta_{helix-pair}(z_i)\Theta_{helix-pair}(z_j)$$
$$\Theta_{helix-pair} = \Theta(z_i, z_m = 15\,\text{Å}) \quad (18)$$

$k_{helix-pair}$ is an a factor for scaling the strength of the helix–pair interaction relative to other terms in the model. For this study, $k_{helix-pair} = 0.5\,\text{kcal/mol}$. The $f(r_{ij})$ in Eq. (18) is a spline-fit potential to the data given in ref. [27] for DMPC lipids and cylindrical inclusions with 5 Å radii. $r_{ij}$ is the distance between the centers of mass of the interacting helices. $\Theta_{helix-pair}(z_i)$ is the switching function given in Eq. (10) with $z_m = 15\,\text{Å}$. $z_i$ and $z_j$ are the distances between the center of mass of helices $i$ and $j$ and the center of the membrane along the membrane normal. The exact form of $f(r_{ij})$ is given in Eq. (19) and is plotted in Fig. 5.

$$f = -3.7673 \times 10^{-6} r_{ij}^5 + 6.0103 \times 10^{-4} r_{ij}^4 - 3.4889 \times 10^{-2} r_{ij}^3$$
$$+ 8.9378 \times 10^{-1} r_{ij}^2 - 9.4119 r_{ij} + 26.745 \quad (19)$$

**Order parameters**. $D$ is the distance between the two termini of GlpG (between the $C_\alpha$ atoms of residues 91 and 271). $Z$ is the the average of the $z$-coordinates of the $C_\alpha$ atoms in GlpG. Unlike $D$, $Z$ is not an experimental observable in the force

spectroscopy experiments of Min et al. Looking at free energy profiles in $(D, Z)$ space allows us to gain new insights into how the changes in these variables are related to each other during folding and unfolding. In particular, we can see whether folding and insertion into the membrane are coupled or whether insertion takes place prior to folding during refolding. In the Supplementary Information, in addition to $D$ and $Z$, three other structural order parameters are used. $Q$ is the fraction of pairwise distances between GlpG's $C_\alpha$ atoms that are within 1 Å of their corresponding value in the crystal structure of GlpG with PDB ID 2XOV. $D_{TM5-6}$ is the end-to-end distance of TM5–6 only. $Z_{TM5-6}$ is the average $z$-value of the $C_\alpha$ atoms within TM5–6 only.

**Molecular dynamics simulations**. Molecular dynamics simulations were run using the LAMMPS molecular dynamics engine[35]. The Langevin integrator in LAMMPS was used with a timestep of 5 fs and a damping time of 10,000 fs. Each simulation was run for 80 million steps. Structures and energies were saved every 4000 steps for further analyses.

**Umbrella sampling**. Since we were interested in comparing the landscape computed using our model with the landscape inferred from the force spectroscopy experiments, we used umbrella sampling along the end-to-end distance to sample GlpG conformations that range from fully folded to fully extended, as well as at end-to-end distances between these two extremes. The large barriers associated with extracting helices out of the bilayer and inserting helices into the bilayer present a challenge to obtaining adequate sampling. To overcome this challenge, we performed temperature replica exchange simulations for all values of the end-to-end distance sampled with umbrella sampling (see Supplementary Figure 8 and Supplementary Note 7).

Umbrella sampling along $D$ was performed by applying two kinds of harmonic potentials to each terminus ($C_\alpha$ atoms of residues 91 and 271) of GlpG. The first is a harmonic potential applied to both termini independently that constrains the termini to remain close to the membrane interface. These potentials have a strength $k = 0.1$ kcal/mol/ $Å^2$, are centered at $z = -17$ Å, and only apply forces in the $z$-direction. The second type of harmonic potential used during umbrella sampling biases $D$, the end-to-end distance, using a strength of $k = 0.02$ kcal/mol/ $Å^2$. Seventy different simulations were run, each with a different biasing center ranging from 40 to 112 Å in increments of 2 Å and 118 to 340 Å in increments of 6 Å.

**Temperature replica exchange**. For each umbrella sampling bias described above, we used temperature replica exchange simulations at 12 temperatures: 300, 335, 373, 417, 465, 519, 579, 645, 720, 803, 896, and 1000 K.

**Free energy calculations**. Using the well-equilibrated set of GlpG conformations at all of the relevant values of the end-to-end distance, we computed free energy landscapes as a function of the end-to-end distance and the average $z$-coordinate, a key order parameter that reflects the extent of insertion into the membrane.

Free energy calculations were performed using the pyMBAR implementation of the MBAR algorithm[36]. The final 20 million steps of each 80 million step trajectory at $T = 373$ K were used as input to the MBAR algorithm. When computing the free energy profiles shown in the Figs. 1, 3, the strength of interactions in $V_{SBM}$ for pairs of residues that were both within residues 91–217 of GlpG was increased by 20%, and the MBAR algorithm was used to determine the free energy profiles using this perturbed energy.

**Folding pathway determination**. By sampling structures along low-free energy pathways between the folded and unfolded states, we obtained a detailed picture of the structural transitions that take place during the force-induced unfolding and spontaneous refolding of GlpG. The folding and unfolding pathways were inferred from the 2D $F(D, Z)$ free energy profiles by manually specifying the starting and ending points and then applying Dijkstra's algorithm for determining the shortest path. The edge weights were obtained from the grid of free energies, $F(D, Z)$, using the formula $w_{edge} = (w_{node_u} + w_{node_v})/2$, where $w_{node_u}$ and $w_{node_v}$ are the weights of two nodes connecting the edge. The weight of each node is given by $w_{node_u} = e^{F(D,Z)}$, where $F(D, Z)$ is free energy at that node. Each node is only connected to its eight nearest neighbors on the rectangular lattice in $(D, Z)$ space.

**Structure selection and alignment**. Structures were chosen at states of interest along the folding and unfolding pathways by selecting several low energy structures from those sampled within the $(D, Z)$ coordinates of interest. Alignment of the selected structures was performed using the CEAlign algorithm as implemented in PyMol[37].

**Calculation of free energy profiles at low and high force**. To obtain free energy profiles at various values of the applied force, an energy term proportional to $D$, the end-to-end distance of GlpG, was subtracted from the energy of all samples obtained using the combined umbrella sampling and temperature replica exchange molecular dynamics simulations described above. The total value of the potential

energy is then given by Eq. (20).

$$V_{total}^{force} = V_{SBM} + V_{bilayer} - k_{force}D, \quad k_{force} \geq 0. \tag{20}$$

The resulting new set of energies, $V_{total}^{force}$, was then used to compute perturbed free energy profiles using the MBAR algorithm as described above. The particular values of the applied force used to obtain free energy profiles in the low-force and high-force regimes were chosen to facilitate the comparison between the computed free energy profiles shown below in the Results section and the free energy profiles drawn in the manuscript by Min et al.[3]. In particular, the value of $k_{force}$ for the low-force regime was chosen such that the lowest free energy nonnative state was approximately 6.5 kT less favorable than the native state basin and the value of $k_{force}$ for the high-force regime was chosen such that the highly extended states were approximately 10 kT more favorable than the native state basin.

## Data availability

Data supporting the findings of this manuscript are available from the corresponding author upon reasonable request. A reporting summary for this Article is available as a Supplementary Information file. The simulation and analysis codes underlying Figs. 1–4 are available online at https://github.com/luwei0917/GlpG_Nature_Communication.

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

## Acknowledgments

We thank Ha Truong for technical assistance and enlightening discussions. This work was supported by Grant R01 GM44557 from the National Institute of General Medical Sciences. Additional support was also provided by the D.R. Bullard-Welch Chair at Rice University, Grant C-0016. We thank the Data Analysis and Visualization Cyberinfrastructure funded by National Science Foundation Grant OCI-0959097.

## Author contributions

W.L., N.P.S. and P.G.W. conceived and designed this study. W.L. and N.P.S. designed the novel aspects of the simulation model. W.L. implemented the novel aspects of the simulation model and performed the simulations. W.L. and N.P.S. performed the data analyses. W.L., N.P.S. and P.G.W. wrote the manuscript.

## Additional information

**Competing interests:** The authors declare no competing interests.

