## [Peer Review File · Nature Communications]

This manuscript has been previously reviewed at another journal that is not operating a transparent peer review scheme. This document only contains reviewer comments and rebuttal letters for versions considered at Nature Communications. Mentions of prior referee reports have been redacted.

Dear Nature Communications Editors –

Thank you for the opportunity to revise our manuscript entitled, “The energy landscape underlying spontaneous insertion and folding of an alpha-helical transmembrane protein into a bilayer” (NCOMMS-18-20648-T). Thanks also to the reviewers for their careful reading of our work and their thoughtful comments. The main text, Online Methods, and Supplementary Information have all been revised and extended. In particular, the Supplementary Information now includes additional new figures, analyses, and explanations that respond to the reviewers’ suggestions. We think that, as a result of answering the concerns of the reviewers, the manuscript has been substantially improved.

Our specific responses to the questions and concerns of each reviewer are given below. The original remarks of the reviewers are shown in black text and our responses are given in blue text. The corresponding changes in the manuscript and methods files have also been highlighted with blue text. The majority of the Supplementary Information is either new or has been revised, so we have chosen not to highlight the changes in the Supplementary Information with blue text.

Reviewers' comments:

Reviewer #1 (Remarks to the Author):

This computational study by Lu, Schafer and Wolynes presents the structural details of the force-induced refolding and unfolding mechanisms of a helical-bundle membrane protein GlpG that have been previously studied experimentally using single-molecule force spectroscopy. This manuscript critically validates the experimental result and further provides new insights into the folding mechanisms of helical membrane protein in the lipid bilayers. This study is an extension of the pioneering effort in the Wolynes group for studying membrane protein folding, a computationally and experimentally challenging problem (2014, 2016 PNAS). Overall, the rationale, results and methodologies are solid and strong, which would be worthwhile to be considered for publication, but there are some major and minor concerns in the discussion part that needs to be addressed for recommendation of acceptance.

We appreciate the reviewer’s positive appraisal of our work. We agree with the reviewer that our work both validates and extends the earlier force spectroscopy work. We have addressed the reviewer’s specific concerns below.

I. Strengths

This is a novel study that enables the crosstalk between experiment and computation, which has been widely carried out for water-soluble protein folding in recent years, but not for membrane protein folding. Therefore, by this study, we expect an exciting synergy between experiment and computation in the future. It is amazing that authors were able to recapitulate the results of the single-molecule study reasonably well. They further provide molecular explanations of several key features in the force-induced folding of GlpG. For example, the difference in the structural ensembles between I1 and N explains the experimentally observed high kinetic barrier in the transition state. These results lead to the reasonable argument that the stability measured by force spectroscopy may be an underestimate. Authors also present a new folding mechanism of GlpG that the insertion and folding of transmembrane segments are coupled and occur in a step-wise manner. Because of the intrinsic kinetic stability of GlpG and the resulting limitation in lowering the force to observe unfolding/refolding, the previous single-molecule study was able to largely detect cooperative unfolding/refolding. Besides the scientific merits, authors improved the methodology by employing a more refined implicit bilayer model implemented with knowledge-based potentials (e.g., inclusion of the residue-burial term, helix orientation term, helix-helix interaction term, etc).

We agree with the reviewer that more conversation between theory and experiment is needed, particularly in the field of membrane protein folding, and we share the reviewer’s enthusiasm for that kind of synergy in this area. We appreciate the reviewer’s recognition of the relevance of our work to the interpretation of the single-molecule experiments on GlpG as well as the novelty of our methodological advances.

II. Concerns

Major concerns

(1) In the membrane protein folding community, a number of researchers regard the two-stage model as a “thermodynamic” model rather than a “mechanistic” model. That is, the insertion is largely “driven” by the hydrophobic effect, while the helix association is driven by the other molecular forces. In that sense, the stepwise insertion/folding “mechanism” suggested by authors as well as the co-translational folding, topology-reorganization and interfacial partition mechanisms mentioned in the discussion would not necessarily contradict with the two-stage model. I suggest authors incorporate this view in the discussion (Section 3.2).

We appreciate the reviewer’s point regarding the two-stage model being interpreted in different ways by different researchers. Any description of a folding mechanism cannot, of course, be in conflict with the two-stage model if one interprets the two-stage model as being purely thermodynamic and not attempting to describe folding mechanisms. We have added clarifying comments into Section 3.3 as suggested by the reviewer. It seems clear to us that, based on the schematic diagrams that are frequently drawn in the literature, some membrane protein scientists do regard the two-stage model as a potentially general model of the kinetic folding mechanisms of membrane proteins in the presence of bilayers. This is why we thought it was useful to point out that, in the first case (the GlpG system) where sufficient experimental and theoretical work has been done to critically evaluate the two-stage model as a description of folding mechanisms, it appears that the two-stage mechanism is not the one that is followed for this system either in vitro or in silico. Of course, we cannot and do not claim that there are no systems for which the two-stage mechanism is operative.

In the last paragraph of Section 3.3, we have therefore inserted the sentence:

“While very useful as a thermodynamic and conceptual model of the way membrane proteins might fold, the validity of the two-stage model as a general kinetic description of the way membrane proteins fold in the presence of bilayers has been largely untested.”

(2) Authors seem to claim the stepwise insertion/folding mechanism as a general principle (I might be wrong) under the force-free condition, supporting the helical hairpin hypothesis. However, the proposed mechanism was derived from the application of force parallel to the bilayer. Thus, by the lateral force, some helices at the bilayer surface (for example, TM5 and TM6 in I2) may be pushed into the water-bilayer interface. Related to this point, it also needs to be clarified what are the effects of the new implicit bilayer model implemented with knowledge-based potentials in comparison to the older bilayer model. Is the aforementioned interfacial partition mainly due to the applied force or the newly implemented potentials?

We agree with the reviewer that the fact that a single membrane protein appears to fold by a helical hairpin insertion mechanism does not constitute a proof of the mechanism’s generality. Although, based on our work and its good agreement with the experimental force spectroscopy observations, we suggest that the two-stage model *may not be* a completely general description of the folding mechanisms for all membrane proteins in the presence of bilayers, we did not say and did not mean to imply that the stepwise helical hairpin insertion mechanism *is*, on the other hand, a completely general universal mechanism. Although there are hints of helical hairpin insertion in the force spectroscopy data collected on membrane proteins other than GlpG, determining the possible generality of the helical hairpin insertion mechanism awaits further experimental and theoretical work, which we have already emphasized at the end of Section 3.5.

We used importance sampling along the end-to-end distance and computed the free energy profiles at various values of the applied force in order to make the comparison of our work to the observations of force spectroscopy as straightforward as possible. At low values of the applied force, the highly extended states that involve more than two helices having been extracted from the membrane are quite high in free energy and are therefore not expected to be significantly populated at equilibrium. In this sense, it is indeed the high force that favors the interfacial partitioning of the transmembrane helices in both our simulations and in the force spectroscopy experiments that have already been carried out in the laboratory. Nevertheless, since the biasing force is applied parallel to the plane of the membrane, the use of the end-to-end distance as a sampling reaction coordinate does not intrinsically favor the interfacial partitioning of transmembrane helices over the separation of helices while remaining within the bilayer as assumed in the two-stage model and, therefore, certainly does not preclude the observation of two stage-like membrane protein folding. The application of force is analogous to most other kinds of perturbations (chemical denaturants, temperature, pressure, etc.) that have been made to protein folding landscapes for the purposes of studying excited states and folding pathways. As is the case when applying other types of perturbations, it is on the basis of our knowledge of the relatively low-lying excited states that we infer the folding pathway starting from an unfolded state when there are no external perturbations.

Clarifying the precise role of each term in the implicit membrane model, although desirable, is non-trivial. Completely repeating the sampling at many values of parameters is not feasible due to the high computational

cost associated with obtaining converged sampling in the presence of the implicit bilayer. Instead, we have computed the expectation value of each energy term in the Hamiltonian along the inferred folding pathway. We have also computed free energy profiles, obtained by perturbation theory, while increasing and decreasing the strength of each term in the implicit membrane model and the structure-based model. These new plots are now shown and discussed in the Supplementary Information.

(3) Overall, some discussion points are weak. Those arguments are based on speculation not strongly supported by experiment. Please see below (minor concerns 4-7).

We have answered the “minor” concerns of the reviewer below and believe that answering these concerns has strengthened the discussion.

Minor concerns

1. p. 6 line 1: “No combination of parameters”

It would be better to specify which parameters have been combined.

We appreciate the reviewer’s preference for specificity when it comes to this point. Unfortunately, a completely systematic search of the parameter space is infeasible due to the considerations mentioned in the answer to the reviewer’s question above. We have modified the sentence mentioned by the reviewer to be as specific as we think it is possible to be without overclaiming.

The revised sentence now reads:

“While developing the current implicit membrane model, all preliminary tests that we performed that involved increasing or decreasing the strength of individual energy terms by between ten and one hundred percent compared to the values given in the Methods section failed to produce a landscape having folded and unfolded states that were completely inserted into the membrane but that were nevertheless approximately equal in free energy and also separated by a significant barrier.”

2. p. 7 lines 9-11: “Therefore, the unfolded state that is reached in the unfolding experiments at high applied force is expected to be much higher in free energy at the lowest forces used by Min et al. to measure unfolding rates than it is at even the highest forces used to measure refolding rates.”

This sentence is complicated and took a while to understand. I recommend a revision.

We have broken up and revised the sentence to improve its understandability.

The revised sentences now read:

“The largest value of the force that was used to measure refolding rates in the study by Min et al. was 7 pN. The lowest value of the force that was used to measure unfolding rates was 12 pN. We see on the basis of our back-of-the-envelope calculation that this seemingly small gap between the two force regimes that were employed gives rise to large changes in the relative free energy of near native states versus highly extended states. In particular, the highly extended states are expected to be quite high in free energy throughout the force range that was used to measure refolding rates, consistent with what we see in our computed free energy profiles.”

3. p. 8 lines 12-16: “The high kinetic barrier separating N and I1, the origin of which was unclear within the two-stage picture of membrane protein folding, is now seen to be associated with the simultaneous insertion and folding of TM5-6 from the bilayer interface in the folding direction and unfolding and extraction from within the bilayer in the unfolding direction.”

This sentence needs to be revised. I recommend,

“the simultaneous insertion and folding of TM5-6 from the bilayer interface in the folding direction and (add “the”) unfolding and extraction from (delete “within”) the bilayer in the unfolding direction.”

We appreciate the reviewer’s suggestion and have changed the sentence accordingly. The sentence now reads:

“The high kinetic barrier separating N and I1, the origin of which was unclear within the two-stage picture of membrane protein folding, is now seen to be associated with the simultaneous insertion and folding of TM5-6 from the bilayer interface in the folding direction and the unfolding and extraction from the bilayer in the unfolding direction.”

4. p.8 Section 3.3: “Amazingly, one of the mechanisms that Booth and Curran put forward in 1999 involves pre-formation of the N-terminal part of Bacteriorhodopsin and a rate-limiting step of cooperative insertion and folding of the two C-terminal helices as a helical hairpin, exactly as we see here for GlpG.” This is a weak argument. Booth and Curran’s suggestion is not based on the solid experimental evidence but on speculation. “Amazingly” could be deleted or changed to “Interestingly”.

We have revised the sentence according to the reviewer’s suggestion by deleting the word “Amazingly”. The sentence now reads:

“One of the mechanisms that Booth and Curran put forward in 1999 involves pre-formation of the N-terminal part of Bacteriorhodopsin and a rate-limiting step of cooperative insertion and folding of the two C-terminal helices as a helical hairpin, exactly as we see here for GlpG.”

5. p.8 Section 3.3: “Recent experiments from the Booth lab, however, do support the ideas that TM1-2 of GlpG rapidly insert into the membrane and that parts of GlpG’s tertiary structure can form co-translationally even in the absence of a translocon, consistent with the mechanism of GlpG’s folding presented in here [15].” Again, the Booth lab’s suggestion is not strongly supported by experimental evidence and purely speculative. The only supporting information is the hydropathy plot of GlpG. In their data, the early accumulation of helical structures could come from any helices, not just from TM1 and TM2.

We appreciate the reviewer’s point regarding the strength and nature of the support provided in the referenced work for the claim that we mention in our manuscript. We have revised the sentence to clarify the nature of the support for that claim in Booth’s work.

The revised sentences now read:

“Booth and coworkers have recently probed co-translational folding of GlpG into membranes in the absence of the translocon using infrared spectroscopy [17], a situation that is somewhat analogous to the situation explored in the force spectroscopy refolding experiments of Min et al. The infrared spectroscopy experiments suggest that, while GlpG is being translated, helices form, these helices insert into the membrane, and some tertiary structure forms. The infrared spectroscopy measurements themselves cannot differentiate between helix formation signals coming from different transmembrane helices, but Booth and coworkers suggest, based on the observation that helices TM1-2 of GlpG are significantly more hydrophobic than other pairs of helices in GlpG, that the first transmembrane helices to insert into the membrane are likely to be TM1-2. These observations and inferences by Booth and coworkers are consistent with the folding mechanism described above in the Results section.”

6. p.9 line 1-2: Our current results suggest that refolding of GlpG would proceed reliably and rapidly from a bilayer interface-associated state to the fully folded state in the presence of a catalyst. I am not sure if this argument would help the overall discussion. The computed barrier of ~7 kT (from I2 to N) does not seem to be high. Therefore, it is difficult to say that a catalyst would be necessary to overcome the barrier.

We agree with the reviewer that it is not clear that a process in the cell involving a 7 kT barrier would require catalysis. We mention in Section 3.1 that our computed barriers are relatively low compared to those that are measured in experiment (this is a common feature of most coarse-grained models of globular protein folding too) and we argue that energetic quantities are more sensitive to model details than are the structural mechanistic aspects of folding, which are the focus of the current manuscript. As we say in the manuscript, obtaining accurate barrier heights and detailed mechanistic understanding of folding processes in the presence of bilayers will probably require cooperation of experiment and theory for the foreseeable future. We have added a clarifying remark to remind the reader that the barrier in the folding direction measured by experiment (~14.6 kT) is about twice as high as the barrier computed for the simulation model (~7 kT).

We have inserted the following sentence immediately after the remark referenced by the reviewer (at the end of the second-to-last paragraph in Section 3.3):

“Whether or not a catalyst would be required to ensure proper folding of GlpG in vivo is a quantitative question having to do with the barrier height that limits refolding ($\approx 15kT$ according to the force spectroscopy experiments) and the rates of other processes taking place in the cell, such as aggregation and degradation, that might interfere with the completion of folding.”

7. p. 9 – p.10, Section 3.5: “The argument about the stability per helix” This is an interesting argument, but TM6 of GlpG is tightly packed against TM6 through the (GXXXGXXXG) motif. Studies by the Urban, Otzen and Hong groups have shown that the packing at this helical interface substantially contribute to GlpG stability.

Authors already argued that the stability of GlpG is underestimated by the previous single molecule study (Section 3.2). Thus, I suggest that authors revise this sentence.

We agree with the reviewer that several lines of experimental work support the importance of the GXXXGXXXG motif in stabilizing the C-terminal part of GlpG. However, in Section 3.2, we argue that the stability inferred by force spectroscopy may be too low as a measure of the total stability of GlpG because the folding transition being probed during refolding by force spectroscopy may correspond to folding only one part of GlpG. Our results give us no reason to believe that the inferred stability of ~ 6.5 kT is too low as a measure of the relative stability of I1 and N in the presence of a bilayer, although we do note in paragraph 2 of Section 3.2 that 6.5 kT is somewhat lower than the stability of helices TM4-6 of GlpG that has been inferred by steric trapping in micelles.

8. Figure 2 and Figure 4

It would be good to add the color codes of individual helices to the figure (Figure 2) and the TM labels in text (Figure 4).

We have added a written description of the colors of the individual helices to the captions of both Figures 2 and 4.

The following sentence has been inserted into the captions of Figures 2 and 4:

“TM1 is red, TM2 is yellow, TM3 is yellow-green, TM4 is green, TM5 is light blue, and TM6 is dark blue.”

We have also added text labels to Figure 4, as suggested by the reviewer, and added the following sentence to the caption:

“The helices are also labeled with text.”

Reviewer #2 (Remarks to the Author):

This paper describes simulations of the folding and unfolding of the membrane protein, GlpG, under force. There has been a dearth of experimental work on membrane protein folding and the situation is even worse when it comes to theoretical work on the global folding of complex membrane proteins like GlpG. The field very much needs the conversation between theory and experiment. Thus, this work is most welcome. GlpG is the only membrane protein whose folding free energy has been examined by three different methods. Moreover, GlpG is one of the few membrane proteins whose folding has been probed by force where the force is applied in the plane of the bilayer. It is therefore a great target for this work as it is easily the most constrained by experiment. The Woynes group has been pioneering simplified energy functions, enabling large scale simulations that can provide mechanistic insight into the folding process and they continue these interesting developments here. The analysis of the folding landscape is quite nice and can provide a roadmap for future simulation work.

We appreciate the reviewer's positive appraisal of our past and present work and share his/her enthusiasm for collaboration between experiment and theory in the field of membrane protein folding.

In addition to the interesting simulation technique developments, the central result of this effort is that the unfolded state in a bilayer involves a C-terminal helical hairpin that is popped out of the bilayer so that folding involves somehow sucking this hairpin into the remaining preformed structure. I find this to be a perfectly plausible hypothesis that will help guide future experiments—exactly what is needed in the field. It is certainly not ruled out by the forced unfolding experiments and I find it striking that Hong's steric trapping experiments are consistent with a more stable N-terminal domain as would be expected from these simulations.

We appreciate the reviewer's positive take on our work and careful comparison to the available experimental data.

MAJOR COMMENTS

1) While I think simulations can provide a possible mechanistic picture where experiments might not yet have tread, they remain hypotheses. There are clearly aspects of the current energy function that don't match physical reality (e.g. the locked helices outside the bilayer). Yet in this paper there are a number of statements that imply that the simulations are the definitive truth. They need to be toned down. The simulations are obviously not a perfect reflection of reality and can't be treated as such.

We agree with the reviewer that no simulations are perfect reflections of reality. We have addressed the reviewers concerns with regards to the phrasing of our statements as described below.

Abstract:

"We find that GlpG, an intramembrane protease folds via sequential insertion of helical hairpins." How about saying "In our simulations we find that..."

The sentence has been revised.

The revised sentence now reads:

"A free energy landscape analysis of the simulations suggests that GlpG, an intramembrane protease, folds via sequential insertion of helical hairpins."

"...are explained by a partially inserted metastable state..." How about saying "...can be explained by..."

The sentence has been changed as suggested by the reviewer.

The revised sentence now reads:

"The low thermodynamic and high kinetic stabilities of GlpG that were recently measured by force spectroscopy can be explained by a partially inserted metastable state that is formed prior to the rate-limiting step of refolding, which leads us to a significant reinterpretation of the rates and stabilities measured by force spectroscopy"

Introduction:

"This reinterpretation of the refolding mechanism substantially changes the meaning of the measured kinetic and thermodynamic stabilities. The rate-limiting step for refolding turns out to be the simultaneous insertion and folding of transmembrane helices 5 and 6 starting from a state with TM1-4 inserted and folded. The thermodynamic stability measured in the force spectroscopy experiments therefore corresponds to the stability of the fully folded state relative to a partially inserted metastable state with transmembrane helices 5 and 6 on the interface of the transmembrane and extramembrane regions." It sounds like you've proven your mechanism, which you haven't.

The sentences have been revised to reflect the need for further experimental work to test the proposed folding mechanism.

The revised sentences now read:

"This reinterpretation of the refolding mechanism, if confirmed by further experimental studies, would substantially change the meaning of the measured kinetic and thermodynamic stabilities. The good agreement between the existing experiments and our calculations suggests that the rate-limiting step for refolding is the simultaneous insertion and folding of transmembrane helices 5 and 6 starting from a state with helices TM1-4 inserted and folded. This structural mechanism of the rate-limiting step implies that the thermodynamic stability inferred by the force spectroscopy experiments may correspond to the stability of the fully folded state relative to a partially inserted metastable state with transmembrane helices 5 and 6 remaining on the bilayer interface."

"The stability of the fully folded state relative to an otherwise unfolded state with complete and correct topological insertion of the transmembrane helices is likely significantly higher than the free energy difference between the folded state and the partially inserted metastable state that was measured by force spectroscopy" How about saying "...may be significantly higher..."

The sentence has been revised as suggested by the reviewer.

The revised sentence now reads:

"The stability of the fully folded state relative to an otherwise unfolded state with complete and correct topological insertion of the transmembrane helices may be significantly higher than the free energy difference between the folded state and the partially inserted metastable state that was measured by force spectroscopy."

Discussion:

"The thermodynamic stability inferred by experiment is, therefore, not the relative stability of the folded and completely unfolded states but, instead, reflects the relative stability of N and I1." How about saying, "...Our simulation results suggest that the thermodynamic stability..."

The sentence has been changed as suggested by the reviewer.

The revised sentence now reads:

"Our simulation results suggest that the thermodynamic stability inferred by experiment is, therefore, not the relative stability of the folded and completely unfolded states but, instead, reflects the relative stability of N and I1."

11.”

“The fact that the lowest free energy non-native state of a transmembrane protein in a bilayer is a partially inserted state has significant implications for our understanding of transmembrane protein evolution, degradation and design.” It’s not a fact.

The sentence has been revised.

The revised sentence now reads:

“The idea that the lowest free energy non-native state of a transmembrane protein in a bilayer is a partially inserted state has significant implications for our understanding of transmembrane protein evolution, degradation and design.”

2) Please define what you mean by “low force” and “high force.” I may have missed it but I couldn’t find it anywhere.

We appreciate the reviewer pointing out the need for clarification on this point. We have added an explanation of the definitions of low force and high force to the Methods section. We have also provided an abbreviated explanation and referenced the detailed explanation in the main text.

The relevant sentences in the main text now read (at the end of the 5th paragraph of the Introduction):

“The particular values of the applied force used to obtain free energy profiles in the low force and high force regimes were chosen to facilitate the comparison between the computed free energy profiles shown below in the Results section and the free energy profiles drawn in the manuscript by Min et al.. By sampling structures along low free energy pathways between the folded and unfolded states, we obtained a detailed picture of the structural transitions that take place during the force-induced unfolding and spontaneous refolding of GlpG. Details of the sampling methods and free energy calculations, including how the free energy landscapes were obtained at various values of the applied force, can be found in the Online Methods.”

The new section that has been added to the Methods now reads:

“9 Calculation of free energy profiles in the low force and high force regimes

To obtain free energy profiles at various values of the applied force, an energy term proportional to D , the end-to-end distance of GlpG, was subtracted from the energy of all samples obtained using the combined umbrella sampling and temperature replica exchange molecular dynamics simulations described above. The total value of the potential energy is then given by Eq. 20.

$V_{\text{forcetotal}} = V_{\text{SBM}} + V_{\text{bilayer}} - k_{\text{force}}D$, $k_{\text{force}} \geq 0$ (20)

This new set of energies, $V_{\text{forcetotal}}$, was then used to compute perturbed free energy profiles using the MBAR algorithm as described above. The value of k_{force} for the low force regime was chosen such that the lowest free energy non-native state was approximately $6.5kT$ less favorable than the native state basin and the value of k_{force} for the high force regime was chosen such that the highly extended states were approximately $10kT$ more favorable than the native state basin to facilitate comparison of the computed free energy profiles with those presented in the manuscript of Min et al. [7].”

3) In the discussion under 3.2, “The discrepancy in the lowest free energy non-native states found at low and high applied force is also evident when comparing the changes in extension during high force unfolding to the sum of the distances to the transition state measured by force spectroscopy.” I don’t understand why this is described as a discrepancy. It’s different and obviously should be different. I’m just not sure what the point is here.

We appreciate the reviewer’s suggestion to clarify these remarks. In making this statement, we are trying to draw the readers attention to the fact that, rather than being a completely unmotivated speculation on our part, a careful reading of the experimental data alone prompts the need for an explanation of the structural differences between the unfolded states favored at high force and at low force. We have revised the wording of this part of the text to clarify our meaning.

The revised sentences now read:

“The fact that the force spectroscopy measurements indicate that there is a large difference in the changes of the end-to-end distance during high force unfolding (> 200 Angstroms) and low force refolding (50 Angstroms) means that there are structural differences between the unfolded state that is favored at high force and the unfolded state that is favored at low force. This fact raises the question, “Is the unfolded state favored at low force simply a more generically compact version of the unfolded state favored at high force, or can a specific partially folded structure account for this difference?” The answer to this question has important implications for the interpretation of the rates measured and stability inferred by force spectroscopy.”

TYPOS

P2 and P7: " force spectroscopy"

P3: "Adopting a native-like orientations"

P9: "transmembrane helices slides"

P9: "avoid become unfolded"

We appreciate the reviewer's careful reading. We have fixed all of the typos mentioned by the reviewer.

Reviewer #3 (Remarks to the Author):

The authors use computational techniques to sample the free energy landscape for the force-dependent unfolding of GlpG in an implicit lipid membrane. The implicit membrane model is a new methodological contribution, and the results suggests a defined unfolding/folding pathway for GlpG in lipid bicelles. The simulation data are compared to experiment force-induced unfolding/refolding results presented in an earlier paper. Although interesting, the contributions made in this paper are insufficient in my opinion to warrant publication in Nature Communications. The free energy surfaces calculated and presented are on their own insufficient to support the proposed broadly relevant new conclusions on the mechanism of membrane protein folding, for reasons discussed in the major comments below.

We appreciate the reviewer's recognition of our methodological advances and the relevance of our work to the interpretation of the experimental force spectroscopy data. We share the reviewer's concern regarding over generalizing based on combined experimental and theoretical analysis of a single protein. For this reason, we had relegated discussing the potential broader implications mentioned by the reviewer to the "Discussion" section of the manuscript, and we have chosen not to include a "Conclusions" section. We have addressed the specific concerns of the reviewer below.

Major comments:

1) The fact that the structure-based interaction potential was increased in TMs1-4 needs to be discussed in the main text, and more carefully shown not to influence the major conclusions. It is concerning that the structure of I1, which has TMs1-4 in the native structure and TMs5-6 non-native could be a simple consequence of these enhanced interactions. The enhanced interactions should also be referred to when using the fact that both the experimental and simulated unfolding pathways start from the C-terminus as evidence for the fidelity of simulation, as this could also be influenced by these altered interactions. The partially folded states mentioned in the online methods that arise when using a uniform potential should be more carefully dissected, including projection on to other collected variables in which they would not overlap the transition state. The presence of these states along the transition path challenge many of the conclusion made in the paper.

We appreciate the reviewer's suggestion to clarify this point. We agree with the reviewer that this aspect of the calculations should be explained in the main text. We have added the following sentences to the beginning of the last paragraph of the introduction:

"To infer folding and unfolding mechanisms and to visualize the free energy landscape both in the low applied force and high applied force regimes, we plotted two-dimensional free energy profiles as a function of the end-to-end distance, D , and the average z -value of the $C\alpha$ atoms, Z . Upon initial examination of the free energy profiles as a function of D and Z , the inferred folding path indicated that folding and insertion would proceed downhill at low values of the applied force, which would be in obvious contradiction to the force spectroscopy measurements. Structural analyses of the basins along the low free energy folding pathway revealed, however, that the apparent lack of a barrier was due to the existence of several near-native ensembles overlapping the transition state in their (D, Z) values. Details of these near-native ensembles and an analysis of the free energy landscape using an alternative set of order parameters can be found in the Supplementary Information (Figs. S3 and S4). Using perturbation theory to preferentially enhance the stability of the Nterminal part of GlpG by 20% helped to clarify the folding pathway in (D,Z) space. We emphasize that the choice of a 20% perturbation is not arbitrary but is a value quantitatively consistent with measurements of subglobal stabilities of GlpG by steric trapping, which indicate that the N-terminal half of GlpG is more stable than the C-terminal half [15]. The resulting energy landscape that includes the 20% perturbation recapitulates all of the major observations of the force spectroscopy study."

We appreciate the reviewer's suggestion to examine the final folding transition using coordinates that do not cause the native-basin structural ensembles to overlap the transition state between I1 and N. We have now included in the Supplementary Information new free energy profiles that allow one to focus on the I1 to N transition. These new free energy profiles show that the relative free energies of I1, N, and the transition state between these states are changed only modestly by the perturbation to the strength of the interactions within the N-terminal part of GlpG and that the apparent barrier heights in the folding and unfolding directions are consistent with those shown in the main text.

The corresponding new text in the Supplementary Information now reads:

"In the main text, D and Z are used as reaction coordinates for plotting the free energy landscapes. D is a natural reaction coordinate to use because it corresponds to the experimental observable in the force spectroscopy experiments of Min et al. [1]. Z is another natural reaction coordinate to use because the extent of burial of the protein into the membrane is arguably the single most important thermodynamic order parameter describing folding and insertion that is not directly an experimental observable in the force spectroscopy experiments. In most cases, D and Z together separate well the low free energy basins ranging from the highly extended states to the folded state. For the near-native states, however, several ensembles in the native basin overlap the transition state for the I1 → N transition, leading to apparent downhill folding at low force that would contradict one of the main experimental observations from force spectroscopy. To clarify the folding pathway in (D, Z) space, the interactions within TM1-4 of GlpG were preferentially stabilized by 20% using perturbation theory. To help us understand the influence of this perturbation on the I1 → N transition, we have plotted the free energy landscapes both with and without applying the perturbation using order parameters that allow us to focus on the I1 → N transition. DTM5-6 is the end-to-end distance of TM5-6 and ZTM5-6 is the average z-value of the Ca atoms within TM5-6.

The structures shown in Section S3 overlap the I1 → N transition state in (D, Z) space. However, because these structures differ from the fully native state only by partial unfolding of TM1-4 and therefore have native-like values of DTM5-6 and ZTM5-6, these ensembles do not overlap the I1 → N transition state in (DTM5-6, ZTM5-6) coordinate space. The free energy profiles shown in Fig. S4 indicate that, both with and without the application of the perturbation, there are two dominant free energy basins corresponding to I1 and N. Furthermore, application of the perturbation does not significantly change the relative free energies of I1, N, and the transition state between I1 and N."

2) The transition state between N and I1 needs to be carefully defined. It is never quite clarified what results in the energetic barrier that would limit GlpG folding, outside of a pretty vague textual description in Section 2.2. If GlpG has to partially unfold, the authors should quantify the degree of unfolding and corresponding energetic cost. If the hydrophilic loop has to pass the hydrophobic lipid membrane at a high energetic cost, the authors should quantify that contribution. If reaction coordinates other than Z and D are appropriate, the authors should replot their analysis in terms of these more informative coordinates. Simulations are not limited to only projections in terms of these experimental observables.

We thank the reviewer for the reminder to clarify the relationship between our choice of global reaction coordinates and the experimental observable, D. We have added a few sentences to the Methods section describing D and Z to emphasize that Z is not an experimental observable.

The new sentences read:

"Unlike D, Z is not an experimental observable in the force spectroscopy experiments of Min et al.. Looking at free energy profiles in (D, Z) space allows us to gain new insights into how the changes in these variables are related to each other during folding and unfolding. In particular, we can see whether folding and insertion into the membrane are coupled or whether insertion takes place prior to folding during refolding."

We agree with the reviewer that it is useful to more fully characterize the structures along the transition from I1 → N. As mentioned above, we have added a table to the Supplementary Information that lists the averages and standard deviations of various collective variables for the structural ensembles highlighted in Figures 2 and 4, including those structures along the transition pathway from I1 → N. We have also computed averaged contact maps for the structural ensembles, which we have added to the Supplementary Information along with a discussion of the contact maps. To help understand the role of the various energetic terms, we have computed both expectation values of each energetic term in the Hamiltonian along the inferred folding pathway as well as obtained perturbed free energy profiles for models where each term in the Hamiltonian has been strengthened and weakened by a small amount using perturbation theory. Finally, as suggested by the reviewer and mentioned above, we have included new free energy profiles that use new reaction coordinates to isolate more clearly the final I1 → N folding transition and have included these free energy profiles in the

Supplementary Information. The reaction coordinates D and Z are preferable for comparison with the experimental results and for looking globally at the folding mechanism, but plotting the free energy landscape as a function of these new more local reaction coordinates shows that stability of I1 compared to the native state and the height of the barrier separating I1 and N are not significantly affected by the perturbation, whose magnitude again we emphasize is consistent with independent experimental measurements by steric trapping.

3) The definition of the low and high force regimes explored in the paper is never presented (that I could find), nor is the methodology that is used to obtain the landscapes in the two different regimes discussed. If an energy landscape was calculated in the absence of a force, and then simply added to the potential created a force, this decision needs to be both explained and justified in terms of the force-pulling experiments that are used for comparison.

Although we did mention in Section 1 of the main text that the free energy profiles at different values of the applied force were obtained by “adding an energy term that is proportional to the end-to-end distance”, we agree with the reviewer that the manuscript would benefit from a more formal discussion of how this was done. We have included a new section in the Methods section that discusses how the free energy profiles at different values of the applied force were computed and how we chose the values of the applied force for the high force and low force regimes.

The new section that has been added to the Methods now reads:

“9 Calculation of free energy profiles in the low force and high force regimes

To obtain free energy profiles at various values of the applied force, an energy term proportional to D, the end-to-end distance of GlpG, was subtracted from the energy of all samples obtained using the combined umbrella sampling and temperature replica exchange molecular dynamics simulations described above. The total value of the potential energy is then given by Eq. 20.

$V_{\text{forcetotal}} = V_{\text{SBM}} + V_{\text{bilayer}} - k_{\text{force}}D, k_{\text{force}} \geq 0$ (20)

This new set of energies, $V_{\text{forcetotal}}$, was then used to compute perturbed free energy profiles using the MBAR algorithm as described above. The value of k_{force} for the low force regime was chosen such that the lowest free energy non-native state was approximately 6.5kT less favorable than the native state basin and the value of k_{force} for the high force regime was chosen such that the highly extended states were approximately 10kT more favorable than the native state basin to facilitate comparison of the computed free energy profiles with those presented in the manuscript of Min et al. [7].”

Fortunately, as discussed by the authors in the original work by Min et al., understanding the perturbations to the energy landscape arising from force is relatively straightforward compared to understanding the perturbations arising from the addition of chemical denaturants. Application of force favors states in proportion to the distance between the points on the molecule where the force is being applied.

In comparing the results of our inferred folding pathway based on equilibrium free energy profiles to the results of the kinetic measurements of Min et al., we are assuming that the equilibrium free energy profiles inferred by Min et al. and presented in their manuscript represent reliable transformations of their kinetic data into equilibrium quantities. Although this process of transformation can be tricky, we have reviewed the assumptions that they used to perform this transformation and have found them to be reasonable. As we discuss in our manuscript, within the two-stage picture of membrane protein folding, it is unclear how such a high kinetic stability could co-exist with such a relatively low thermodynamic stability. However, the helical hairpin insertion folding mechanism described in our manuscript provides a natural explanation for both of these features and, in turn, supports the reasonability of the transformation of the kinetic data to equilibrium quantities performed by Min et al..

4) The arguments made at the end of section 3.3 in the discussion, while interesting and not unreasonable, are not strongly supported by the data presented in the paper. Either the link between the results and the statements should be described more clearly, or the conclusions should be made less strongly.

In Section 3.3, we used the phrases “*Our current results suggest that...*” and “*The present simulations along with their harmony with experiment suggest that...*”, which we believe to be accurate and do not seem particularly strong to us. Furthermore, we ended the section by noting that determining whether or not the *in vivo* folding mechanism of GlpG follows a two-stage model “*will require further experiments and simulations.*” We hope by these phrasings that readers will understand the relative strengths and weaknesses of the ideas in the paper.

In section 3.5, it is difficult to make conclusions about other membrane proteins which have not been simulated, because, as the authors explain well, the molecular mechanism underlying the experimentally obtained force profiles is often ambiguous.

We agree with the reviewer that it is difficult to draw conclusions about those other proteins, which is why we have avoided doing so, instead choosing to include these remarks on other systems only in the “Discussion” section. Nonetheless, we think that discussing these points may well be useful to both experimentalists and theorists looking to carry out further studies on those systems.

Minor comments:

1) The introduction discusses phi-values and backtracking without defining these concepts, which would prevent the unfamiliar reader from understanding the points being made.

We appreciate the reviewer’s suggestion to make this part of the manuscript accessible to a wider audience. We have added as much explanation of the relevant concepts as we believe that we can do without breaking the flow of the logic of the manuscript. We have furthermore added citations that interested readers can follow to learn more about these concepts.

The revised sentences now read:

“The same structure-based forcefield used in the present study was previously used in a prior study to elucidate the origin of the puzzling preponderance of GlpG’s negative ϕ -values [9, 2] when GlpG is folded in mixed detergent micelles [10, 8]. Refolding in micelles turns out to involve “backtracking” [11], which gives rise to negative ϕ -values. ϕ -values are measured experimentally by comparing the change in the apparent stability of the transition state (by measuring changes in folding rates) to the change in the stability of the folded state upon making a mutation. GlpG’s negative ϕ -values arise from mutations that both decrease the stability of the folded state and increase the folding rate by allowing more facile backtracking. In [8], it was shown that the large number of negative ϕ -values found for GlpG folding in micelles could be attributed to a multi-step folding mechanism that involves breaking and eventually reforming an interface while proceeding in the folding direction (backtracking). This folding complexity was also shown to be partially attributable to GlpG’s modular structure and also to the high degree of conformational entropy in the micellar unfolded state.”

2) The authors discussed results that demonstrate the robustness of their conclusions to the choice of parameters in their potential. It would be nice to see the data for these controls presented in a figure, perhaps in the SI.

In Section 3.1 we state that we expect energetic aspects of the computed free energy landscapes to be less robust than structural/geometric aspects of the inferred folding mechanisms. We furthermore admit that we currently lack the experimental data that would be necessary to optimally parameterize the implicit membrane potential. As discussed above, we have computed expectation values of the various terms in the Hamiltonian as well as perturbed free energy surfaces obtained while changing the strengths of the parameters in the Hamiltonian. These new figures are now included in the Supplementary Information.

3) The convergence of the umbrella sampling, although discussed, should be shown.

We have added a new section to the Supplementary Information that now contains a figure indicating that the sampling used to compute the free energy profiles presented in the main text has converged.

4) The selection of structures presented in Figs. 2 and 4 and described in section 8 should provide more detail. It would be interesting to define how representative any given conformation at a particular D and Z values is in terms of alternative collective variables

We agree with the reviewer that it is useful to more fully characterize these structural ensembles highlighted in Figures 2 and 4. As mentioned above, we have added a table to the Supplementary Information that gives the average values and standard deviations of structural collective variables and average contact maps for each of the structural ensembles discussed in Section 2.

5) A figure showing the experimental setup, perhaps a simplified version of Fig. 1a in Ref. 3, might aid the reader in understanding the goals of the simulation.

We thank the reviewer for this suggestion. We have added a schematic figure, adapted from the manuscript of Min et al., to the Supplementary information.

6) A discussion of why the low-force pulling landscape has no unfolded state minimum could be useful. Is this expected based on experimental results?

The experimental results imply the presence of a relatively low-lying (~6.5 kT) non-native state that is separated from the native state by a large barrier (~21 kT in the unfolding direction). We believe that the presence of I1 on the landscape provides an explanation for both of these findings, and this aspect of our work

is discussed extensively throughout the manuscript. Both the relative free energy of the native and completely unfolded states and the structure of the lowest free energy non-native state are not easily inferred from the experimental data alone, and we have discussed this in Section 3.2.

7) The numerical value of z_m should be provided to aid reproducibility

We thank the reviewer for pointing out that this information was missing from a few places in the Methods section. The value of z_m used for each term in the implicit membrane is now given explicitly in each section.

Reviewers' Comments:

Reviewer #1:

Remarks to the Author:

I thank authors for carefully and rigorously addressing my concerns. The clarity and quality of the manuscript have been substantially improved to a reasonable level for recommending an acceptance. New supplementary figures showing the free energy landscapes under various parametrized conditions (response to other reviewers) are persuasive and will help readers understand the value of this manuscript. This work is a tour de force!

Only minor points:

Figure S4 bottom panel: y-axis label is missing.

Figure S5: The scale of heat map and the x- and y-axis labels are missing.

Reviewer #2:

Remarks to the Author:

The authors have done a thorough job of responding to the review points and I strongly support publication of this pioneering work.

Reviewer #3:

Remarks to the Author:

The manuscript has been revised as requested to clarify various technical aspects. My central concern remains largely unchanged that the "free energy surfaces calculated and presented are on their own insufficient to support the proposed broadly relevant new conclusions on the mechanism of membrane protein folding." This has been somewhat addressed by the authors by softening the wording of the speculation in the discussion section.

Dear Nature Communications Editors –

Thank you for the opportunity to revise our manuscript entitled, “Energy landscape underlying spontaneous insertion and folding of an alpha-helical transmembrane protein into a bilayer”. During this latest revision, the scientific contents of the manuscript have not been changed. The manuscript has, however, been fully revised to comply with the requirements for publication in Nature Communications. Below, we have included a point-by-point response to the reviewer’s comments. Our responses to each point are shown in blue.

Reviewer’s comments:

Reviewer #1 (Remarks to the Author):

I thank authors for carefully and rigorously addressing my concerns. The clarity and quality of the manuscript have been substantially improved to a reasonable level for recommending an acceptance. New supplementary figures showing the free energy landscapes under various parametrized conditions (response to other reviewers) are persuasive and will help readers understand the value of this manuscript. This work is a tour de force!

We appreciate the reviewer’s helpful comments and positive appraisal of our work.

Only minor points:

Figure S4 bottom panel: y-axis label is missing.

We have added a label to the x-axis (the y-axis label was already present).

Figure S5: The scale of heat map and the x- and y-axis labels are missing.

We have added labels for the x- and y-axes as well as the color scale.

Reviewer #2 (Remarks to the Author):

The authors have done a thorough job of responding to the review points and I strongly support publication of this pioneering work.

We appreciate the reviewer’s helpful comments and positive appraisal of our work.

Reviewer #3 (Remarks to the Author):

The manuscript has been revised as requested to clarify various technical aspects. My central concern remains largely unchanged that the free energy surfaces calculated and presented are on their own insufficient to support the proposed broadly relevant new conclusions on the mechanism of membrane protein folding. This has been somewhat addressed by the authors by softening the wording of the speculation in the discussion section.

We appreciate the reviewer’s helpful comments. We believe that the good agreement between our theoretical analyses and the previously reported single molecule experiments provides strong support for idea that GlpG folds

according to the folding mechanism described in the manuscript. We agree with the reviewer that the generality of this mechanism remains an open question, and we look forward to future experimental and theoretical work in this area.

Best regards,

Wei Lu, Nicholas P. Schafer, and Peter G. Wolynes